# Telomerase-Targeted Cancer Immunotherapy

**DOI:** 10.3390/ijms20081823

**Published:** 2019-04-12

**Authors:** Eishiro Mizukoshi, Shuichi Kaneko

**Affiliations:** Department of Gastroenterology, Graduate School of Medicine, Kanazawa University, Kanazawa City, Ishikawa 920-8641, Japan; skaneko@m-kanazawa.jp

**Keywords:** hTERT, T cell, peptide vaccine, dendritic cell, chimeric antigen receptor, immunotherapy

## Abstract

Telomerase, an enzyme responsible for the synthesis of telomeres, is activated in many cancer cells and is involved in the maintenance of telomeres. The activity of telomerase allows cancer cells to replicate and proliferate in an uncontrolled manner, to infiltrate tissue, and to metastasize to distant organs. Studies to date have examined the mechanisms involved in the survival of cancer cells as targets for cancer therapeutics. These efforts led to the development of telomerase inhibitors as anticancer drugs, drugs targeting telomere DNA, viral vectors carrying a promoter for human telomerase reverse transcriptase (hTERT) genome, and immunotherapy targeting hTERT. Among these novel therapeutics, this review focuses on immunotherapy targeting hTERT and discusses the current evidence and future perspectives.

## 1. Introduction

For the survival of multicellular organisms, such as humans, the homeostasis and function of the cellular network that forms different organs needs to be maintained. Cancer is an aggregate of abnormal cells that arise from mutations. These cells impair the normal cellular network by replicating and proliferating in an uncontrolled manner, infiltrating into tissues, and metastasizing to distant organs, eventually leading to the death of the organism. One of the mechanisms that enables cancer cells to behave in such a manner is controlled by telomeres and telomerases [1,2].

DNA replication is required for cells to divide and proliferate. During a normal cycle mediated by DNA polymerase, sections of telomeres found on both ends of a DNA strand are not fully replicated. As a result, telomeres are gradually lost with each cycle. Telomeres protect the ends of chromosomes, and the shortening of telomeres results in the exposure of the ends of DNA strands. As a result, DNA replication no longer occurs, and the cell stops replicating and eventually dies. In our body, this mechanism prevents cells with an abnormal proliferative capacity to become cancerous. On the other hand, many cancer cells are able to maintain telomeres because they have an activated form of telomerase that is involved in the synthesis of telomeres. Thus, as discussed above, telomerases help cancer cells to divide infinitely and to act in an uncontrolled manner [3].

Studies to date examined the mechanisms underlying the survival of cancer cells as targets for cancer therapeutics [4,5,6,7]. These efforts led to the development of telomerase inhibitors as anticancer drugs, as well as drugs that target telomere DNA, viral vectors that carry a promoter for the human telomerase reverse transcriptase (hTERT) genome, and immunotherapy strategies that target hTERT. In addition, recent studies have shown that mutations exist frequently in the promoter region of the hTERT gene and increase the activity of telomerase to play a major role in tumorigenesis [8,9]. These findings have triggered researchers to reconsider the usefulness of treatments targeting hTERT [10]. This review focuses on immunotherapy targeting hTERT and discusses the current evidence and future perspectives.

## 2. Recent Advancements in Cancer Immunotherapy

Cancer immunotherapy is considered the fourth pillar of cancer treatment after surgery, chemotherapy, and radiation therapy and includes the use of cytokines, antibodies, checkpoint inhibitors, and immune cells such as dendritic cells (DCs) and T cells. Among the different types of cancer immunotherapy, checkpoint inhibitors and chimeric antigen receptor T-cell (CAR-T) therapy have been used successfully for the treatment of cancer in clinical settings [11,12,13,14]. Unlike conventional methods that target cancer cells, these types of cancer immunotherapy strategies are novel as they target the host immune system and may bring a paradigm shift in the treatment of cancer. Based on the success of these cancer immunotherapy strategies in the clinical setting, cancer and immunology studies have reemphasized the importance of the role of T cells to recognize tumor antigens and to subsequently eliminate cancer cells [15,16].

## 3. Expression of hTERT as a Target Antigen in Cancer Immunotherapy

The first step in the development of cancer immunotherapy targeting tumor-specific immune responses is to identify target tumor-associated antigens (TAAs). To date, several TAAs including cancer/testis antigen and carcinoembryonic antigen (CEA) have been identified and targeted in immunotherapy [17,18]. Once a cancer cell dies by mechanisms such as apoptosis, a part of the cell is excreted into its surroundings and is internalized by endosomes of dendritic cells (DCs). Upon internalization, TAAs are degraded and form long peptides composed of 10–20 amino acids, which then bind to the major histocompatibility complex (MHC) class II molecule to be expressed on the cell surface. A recognition of these peptides by the T-cell receptor (TCR) of CD4^+^ helper T (Th) cells results in the activation of Th cells. DCs also export TAAs to the cytoplasm where the antigens are degraded by proteasomes to form short peptides composed of 9–11 amino acids. Once these peptides bind to MHC class I molecules and are expressed on the cell surface, they are recognized by the TCR of CD8^+^ cytotoxic T lymphocytes (CTLs) to activate CTLs [19]. CTLs are important effector cells that kill cancer cells (Figure 1).

Techniques for identifying novel TAAs include serological analysis of expression cDNA libraries (SEREX) and cDNA microarrays. After the amino acid sequences of TAAs are identified, numerous short peptides that induce MHC class I-restricted CTLs can be identified using algorithm analyses and transgenic mouse models expressing human leukocyte antigen (HLA) [20,21].

hTERT is one such TAA and is overexpressed in over 85% of tumors, including tumors of hematopoietic tissues and solid tumors [22]. Its expression in normal cells is limited to testicular cells, hematopoietic stem cells, basal keratinocytes, and activated lymphocytes [23,24]. It is also overexpressed in cancer stem cells, in which hTERT plays a role in the replicative features and immortality of the cells [25,26]. hTERT is, therefore, an attractive target for cancer immunotherapy, including as a method to target cancer stem cells. Studies to date demonstrated that short and long peptides originating from hTERT form complexes with MHC class I and class II molecules; these complexes are then expressed on the cell surface to elicit CTL and Th cell responses in vitro [27,28,29]. As hTERT is immunogenic, it is considered a universal TAA that can be used as a target to elicit antitumor immunity. Indeed, several hTERT-derived peptides have been identified as targets for cancer immunotherapy (Table 1) [29,30,31,32,33,34,35,36,37,38,39,40,41,42,43,44].

## 4. Development of Peptide Vaccines That Target hTERT

The safety profile, immune response, and antitumor effects of several vaccines using hTERT-derived peptides have been evaluated for a number of cancer types. The majority of these vaccines are highly specific to tumors [45] and possess both MHC I and MHC II epitopes within their amino acid sequences [46]. Currently, over 30 hTERT peptides are being used as mimotopes, which mimic the structure of epitopes. Although many of them are expressed on MHC class I molecules and play a role in enhancing CTL response, some are expressed on MHC class II molecules and function in inducing Th cell responses [29,42]. Vaccines based on hTERT-derived epitopes that have been described to date are listed in Table 2. They include several drugs, such as GV1001 (used in combination with the granulocyte-macrophage colony-stimulating factor (GM-CSF)), GX301, GRNVAC1, and VX-001 [6,47,48,49,50,51], which are described in the following sections.

### 4.1. GV1001

GV1001 is an MHC class II-restricted peptide vaccine composed of 16 amino acids from the active site of hTERT (611–626, EARPALLTSRLRFIPK) [69]. It requires GM-CSF or toll-like receptor 7 (TLR-7) as an adjuvant and elicits potent CD4^+^ and CD8^+^ T-cell responses, as well as CTL activation [57]. In addition to the immunological mechanisms, previous studies demonstrated that GV1001 acts on cells directly. Specifically, GV1001 penetrates through the cell membrane and localizes in the cytoplasm, where it reduces the level of heat shock protein (HSP) inside the cell and on the cell surface. Furthermore, it reduces the expression of HSP90, HSP70, hypoxia-inducible factor (HIF)-1α, and vascular endothelial growth factor (VEGF) in tumors under hypoxic conditions. GV1001 was suggested to have high antitumor effects because it is a cell-penetrating peptide [70,71]. In renal cell carcinoma, GV1001 effectively induced the apoptosis of cancer cells by reducing angiogenesis [72]. Moreover, GV1001 was demonstrated to have anti-inflammatory effects [73,74,75] and antiviral activity [76,77], in addition to protective effects from β-amyloid-induced neurotoxicity in the central nervous system [78].

GV1001 is the most advanced vaccine among all vaccines that target hTERT and was the first vaccine to be examined in non-randomized clinical trials. It has been tested as a vaccine to treat different types of cancers such as advanced pancreatic cancer, non-small cell lung cancer (NSCLC), and melanoma [55]. Clinical trials to date reported the following. First, a phase I/II study demonstrated that GV1001 induced specific T-cell responses in 50–80% of advanced pancreatic cancer and lung cancer patients with no clinical toxicity [79]. A study with lung cancer patients further found that the administration of GV1001 resulted in both CTL and CD4^+^ T-cell responses [54]. One of the advantages of this vaccine is that it can be administered without the HLA-typing of patients; thus, it is optimal as a universal cancer vaccine.

The efficacy of GV1001 has also been confirmed when used in combination with other peptide vaccines. A phase I/II study demonstrated that the combination of GV1001 with the HLA-A2-restricted CTL epitope for telomerase (HR2822; hTERT_540–548_) elicited an immune response with an excellent safety profile in patients with NSCLC [57]. In this trial, an immune response was induced in 86% of the patients (12/14), with one patient achieving a complete response (CR). The study also examined its mechanism of action and revealed that GV1001-specific Th cells recognized antigen-presenting cells (APCs) that had internalized cancer cells in the tumor and lymph nodes. Furthermore, the vaccine did not affect bone marrow cells. Other clinical trials examined the combinations of GV1001 with GM-CSF, temozolomide, and p540 peptide [52,53,55,56,58]. A study on patients with pancreatic ductal carcinoma reported that the combination of GV1001 with gemcitabine led to tumor cell death and a significant loss of fibrous tissue within tumors [80]. However, in a phase III trial for patients with metastatic pancreatic cancer, the addition of GV1001 to chemotherapy (gemcitabine and capecitabine) did not improve the overall survival [56,81].

### 4.2. GX301

GX301 is a vaccine consisting of 4 hTERT-derived peptides (hTERT_540–548_, hTERT_611–626_, hTERT_672–686_, and hTERT_766–780_). It is able to bind to both MHC class I and II molecules and contains montanide ISA-51 and imiquimod as adjuvants [61]. In a phase I study on patients with stage IV prostate cancer and kidney cancer, all patients exhibited immune responses to at least one of the peptides. This study suggested that multi-peptide vaccines are more effective as they enhance the immune response in a greater number of responders than single-peptide vaccines [82].

### 4.3. UV1

UV1 is a second-generation telomerase peptide vaccine that was described recently. It consists of the 3 most common hTERT-derived peptides that are found in long-term cancer survivors, namely hTERT_691–705_ (RTFVLRVRAQDPPPE), hTERT_660–689_ (ALFSVLNYERARRPGLLGASVLGLDDIHRA), and hTERT_652–665_ (AERLTSRVKALFSVL). The vaccine was tested in a phase I/IIa trial for patients with metastatic hormone-naïve prostate cancer and was administered as an immunomodulator with GM-CSF for 6 months. The vaccine elicited an immune response in 85.7% of the patients (17/21) and reduced the level of prostate-specific antigen (PSA) in 64% of the patients. Magnetic resonance imaging (MRI) further demonstrated the disappearance of tumors in the prostate in 45% of the patients after vaccination. Most adverse events associated with the vaccine were classified as grade I [59].

### 4.4. Vx-001

Vx-001 is a vaccine consisting of 2 peptides: hTERT-derived low-affinity cryptic hTERT peptide (RLFFYRKSV) and its optimized mutant hTERT peptide (YLFFYRKSV). The latter has an enhanced affinity to MHC class I molecules because the first amino acid was replaced with a tyrosine residue [83,84]. The antitumor efficacy of Vx-001 has been demonstrated in phase I/II clinical trials for different types of cancers such as NSCLC, melanoma, breast cancer, and bile duct cancer. Furthermore, in these trials, the vaccine elicited a strong hTERT-specific immune response, had a good tolerance profile, induced only mild side effects, and improved clinical outcomes [60,84].

## 5. Immunotherapy Using hTERT-Targeting Dendritic Cells (DCs)

DCs are the most potent antigen-presenting cells in the body and play an important role in inducing adaptive immunity and supporting the innate immune response. Over the past decade, DCs have been used as a tool to induce potent antitumor immune responses in cancer immunotherapy [85]. In the United States, a DC vaccine called sipuleucel-T was approved by the Food and Drug Administration (FDA) to be used for patients with metastatic prostate cancer. Sipuleucel-T is a cell product that was developed by culturing DCs with a tumor antigen (prostatic acid phosphatase (PAP) fusion protein) and was found to prolong survival by approximately 4 months in a phase III trial [86]. The use of DCs has also been investigated to develop hTERT-targeted immunotherapy (Table 2) [63,65].

### 5.1. GRNVAC1

GRNVAC1 is a DC-based cancer vaccine produced by transducing mature DCs from patients with mRNA encoding hTERT and lysosomal associated membrane protein (LAMP) 1 [64]. LAMP1 brings hTERT into lysosomes where it is degraded into small peptides. Antigen epitopes that are presented by DCs after the administration of the vaccine represent different sections of the hTERT peptide to elicit polyclonal immune responses [69,87]. Clinical studies to date have reported that GRNVAC1 is safe and well-tolerated [79]. A study on metastatic prostate cancer patients examined the effects of DCs that were transfected with mRNA encoding chimera LAMP1 and hTERT. GRNVAC1 did not elicit autoimmune responses, and multiple administrations of the vaccine were well-tolerated by patients. In addition, the vaccine induced immune responses via antigen-specific CD8^+^ and CD4^+^ T cells in the patient population [64]. A long-term administration of the vaccine has also been reported to be effective for patients with acute myeloid leukemia [66]. GRNVAC2, another DC-based vaccine, is produced in the same way as GRNVAC1, except that it originates from human embryonic stem cells instead of leukapheresis. GRNVAC2 may be a better option in terms of the delivery system [79,83]. These vaccines may be more advantageous than peptide vaccines as they are not restricted by HLA and may be effective against tumors with unknown T-cell epitopes.

Additional studies examined varying methods to administer DC-based hTERT vaccines. For example, one study examined the effects of an indoleamine 2,3-dioxygenase (IDO)-silenced DC vaccine that was simultaneously transfected with mRNA encoding survivin and the hTERT tumor antigen. In this study, the vaccine induced T-cell responses to survivin and hTERT in patients with metastatic melanoma who were pretreated with ipilimumab. Furthermore, T-cell responses against the melanoma-associated antigen recognized by T cells (MART-1) and NY-ESO-1 were detected in the peripheral blood. Patients who underwent the treatment had fewer metastases to the lung, liver, and skin and had an improved overall performance status [88]. Another study examined a technique to use a recombinant adenovirus encoding hTERT cDNA to transfect DCs. DCs produced by this method were used to give rise to hTERT-specific CTLs from autologous T cells in vitro. This technique resulted in the expression of the antigen and improved CTL response [89].

### 5.2. TAPCells

Another approach using DCs is the production of therapeutic dendritic-like cells called tumor antigen presenting cells (TAPCells). The TAPCell-based vaccine was evaluated for over 120 patients with stage III and IV melanoma and for 20 patients with castration-resistant prostate cancer in phase I and phase I/II studies. In these studies, the vaccine increased the survival rate of melanoma patients, prolonged the doubling time of PSA, and elicited T-cell responses in prostate cancer patients. Furthermore, over 60% of the patients had delayed-type hypersensitivity (DTH) reactions against the lysates. This suggested that the treatment promoted antitumor immune memory, which is associated with the clinical efficacy. The study also demonstrated that the TAPCell-based vaccine increased the number of Th1 and Th17 cells and that the addition of Concholepas concholepas hemocyanin (CCH) as an adjuvant was safe and further enhanced the immune response [67].

### 5.3. Other DC-Based Approaches

Mehrotra et al. recently performed a phase I trial to examine the use of DCs that had been pulsed with 3 different A2-restricted peptides. A pulsed DC vaccine was generated by hTERT (TERT572Y), CEA (Cap1-6D), and survivin and was evaluated for the treatment of pancreatic cancer. The treatment elicited specific T-cell responses, and stable disease (SD) was achieved in 50% of the patients. The medial overall survival was 7.7 months. The vaccine was well-tolerated, with the most common side effects being transient fatigue and flu-like symptoms [68].

As mentioned above, severe adverse events have not been observed in hTERT-targeted immunotherapy. However, in this immunotherapy, it might be necessary to be careful about abnormalities in the host’s immune system. Hematopoietic progenitor cells and both B and T lymphocytes have a high telomerase activity. This means that hTERT-based anticancer immunotherapy not only kill cancer cells but also these lymphocytes with a high telomerase activity. In previous studies with hTERT-derived peptide vaccine and dendritic cells, no serious adverse events regarding lymphopenia have been reported (Table 2). However, it should be carefully observed in future studies using new treatment strategies described later.

## 6. DNA Vaccines

### 6.1. phTERT

Recombinant DNA techniques can be used with genomes encoding the hTERT peptide to improve the efficacy of epitope presentation to T cells. Plasmids containing these genomes can be delivered to antigen-presenting cells by a gene gun or electroporation. Compared with peptide-based vaccines, DNA-based vaccines are more cost-effective. phTERT is a full-length vaccine optimized and synthesized as a DNA vaccine that encodes hTERT. When administered to mice and nonhuman primates by electroporation, phTERT induces potent and broad hTERT-specific CD8^+^ T-cell responses, including T cells expressing CD107a, IFN-γ, and TNF-α. Moreover, significant IFN-γ responses and a release of antigen-specific perforin were observed in immunized monkeys, suggesting that phTERT overcomes immune tolerance and elicits a potent cell cytotoxicity in the in vivo model of human immunology. Furthermore, one previous study used an HPV-related tumor model to examine the preventive and therapeutic potential of the phTERT vaccine and found that the vaccine slowed the tumor proliferation rate and improved survival [90].

### 6.2. INVAC-1

INVAC-1 is an optimized plasmid encoding an inactive form of hTERT that can be administered via electroporation-based intradermal injection. In a mouse model, INVAC-1 induced hTERT-specific T-cell responses, including CD4^+^ Th1 effector and memory CD8^+^ T cells. In the melanoma model, INVAC-1 resulted in a survival rate of 50%, as well as a significant tumor growth delay compared with the control group [91].

## 7. Cell-Based Immunological Approaches

Cell-based approaches include the use of human umbilical vein endothelial cells (HUVECs) immortalized with hTERT genes by lentiviral infection. Cells produced by this technique have a high telomere activity and express CD31, VEGF receptor-2 (VEGFR-2), and integrin α5. In one previous study, these cells were irradiated to terminate cell proliferation and injected subcutaneously as a vaccine into mouse models of lung cancer and colorectal cancer. The vaccine elicited both humoral and cell-mediated immune responses, suggesting that it has both protective and therapeutic antitumor effects [92].

Another approach demonstrated the use of adenovirus as a vector. With this technique, a mixed vaccine was developed using a mannan-modified adenovirus that expressed hTERT and VEGFR-2. The vaccine elicited potent antitumor immune responses and inhibited intertumoral angiogenesis by activating CTLs reactive to hTERT and VEGFR-2 [93].

## 8. Gene-Modified T-Cell Therapy

Gene-modified T-cell therapy has been developed as a method to deliver T cells that are specific for different types of cancers. It uses T cells that are genetically engineered to produce TCRs that recognize tumor antigens and their epitopes [94,95]. Currently, there are two methods for developing gene-modified T cells; one is based on the use of tumor antigen-specific TCRs originating from tumor-specific T cells or their clones [96,97], and the other is based on the use of a chimeric antigen receptor (CAR) [13,98,99]. The extracellular portion of the CAR is a single-chain antigen recognition receptor composed of the variable regions of heavy and light chains of a monoclonal antibody specific to the tumor surface antigen, and the intracellular portion of the CAR is created by a binding of co-stimulatory molecules to the intracellular portion of the TCR.

TCR-engineered T (TCR-T) cells are produced by modifying T cells with the genome of TCRs that specifically recognize the complex of tumor-surface antigen peptides and major histocompatibility complex (MHC) molecules. Thus, TCR-T therapy is only effective if tumor cells express the target antigen epitopes and MHC molecules. As discussed above, previous studies demonstrated that many cancer cells express epitopes originating from hTERT. Thus, TCR-T-cell-based immunotherapy targeting these epitopes may be effective against tumor cells expressing hTERT. Studies to date identified TCRs for hTERT and have suggested their use for immunotherapy [62,100,101,102].

## 9. hTERT-Targeted Cancer Immunotherapy: Future Perspectives

Many immunotherapies using hTERT-derived peptides, DNAs, and DCs have been developed, but their effects so far are modest. The reason is that hTERT is a self-antigen, and T cells exerting an effect against such self-antigens are hard to induce in vivo. In addition, the TCR affinities of the induced T cells are low, and therefore, the antitumor effect of these T cells might be weak. Furthermore, the expression of antitumor effects by these vaccines usually need some time, and such an antitelomerase therapy may favor the emergence of adaptive responses, such as the activation of the alternative lengthening of telomeres mechanism reported by Hu et al. [103]. In order to overcome such a point, it is important to administer a large amount of T cells having a TCR capable of exerting a certain antitumor effect.

As described above, in the field of cancer immunotherapy, treatment with TCR gene-modified T cells has been attempted in many cancer types. In immunotherapy targeting hTERT developed so far, this therapeutic method using genetically modified T cells is a mechanism to administer T cells with TCRs that can reliably exert antitumor effects in vivo and it is the most promising treatment in that a reliable antitumor effect can be obtained in a short time. On the other hand, since it is thought that an immunosuppression mechanism via immune checkpoint molecules such as PD-1 is considered to be working at the tumor locality, a combined therapy with immune checkpoint inhibitors will be further promoted and seems to be effective.

As research on cancer immunotherapy resulted in the wide use of checkpoint inhibitors in clinical practice, several important factors have been reemphasized. First, the importance of the role of a host immune system, particularly of T cells, to recognize tumor antigens and to subsequently eliminate cancer cells was suggested. Second, checkpoint inhibitors may not be as effective for patients who lack T cells that are able to recognize tumor antigens because the cells have not been induced or have not infiltrated the tumors. These patients will require an induction or direct administration of tumor-specific T cells in order for the cells to reach the tumors [104]. Recent studies suggested that neoantigens, which arise as a result of genetic mutations in the tumor, can effectively eliminate tumor cells [105,106]. However, neoantigens are not suited for use as universal antigens because they differ among patients and cancer types. On the other hand, hTERT is expressed in most cancer cells and may, therefore, be more advantageous as a target antigen for cancer immunotherapy. Furthermore, one type of immunotherapy alone (e.g., anti-PD-1 antibody or anti-CTLA-1 antibody) is not effective for the treatment of solid tumors [107,108,109]. As such, recent studies have focused on the development of combination treatments such as those with checkpoint inhibitors and molecular-targeted agents [12,110,111,112]. The efficacy of treatments that induce hTERT-specific responses may also improve if they are combined with checkpoint inhibitors, molecular-targeted agents, and/or other types of immunotherapy. Therefore, future investigations should focus on the development of complex strategies incorporating hTERT-specific immunotherapy.

## 10. Conclusions

Among numerous tumor antigens, telomerase is an attractive target for cancer immunotherapy because hTERT is a universal antigen and its expression is specific to cancer cells. As such, many studies to date focused on the development of telomerase-targeted strategies. Studies on antitumor immunity have advanced rapidly in recent years, and many of the findings are currently being applied clinically. However, recent evidence also suggests that a single type of immunotherapy is insufficient to eliminate solid tumors. Thus, there is a need to develop novel strategies that combine different types of immunotherapy, molecular-targeted agents, and chemotherapy in order to improve the prognosis of cancer patients. hTERT-targeted immunotherapy is no exception in the development of multiplex immunotherapy strategies, and these efforts should advance the future of cancer treatment.

## Figures and Tables

**Figure 1 ijms-20-01823-f001:**
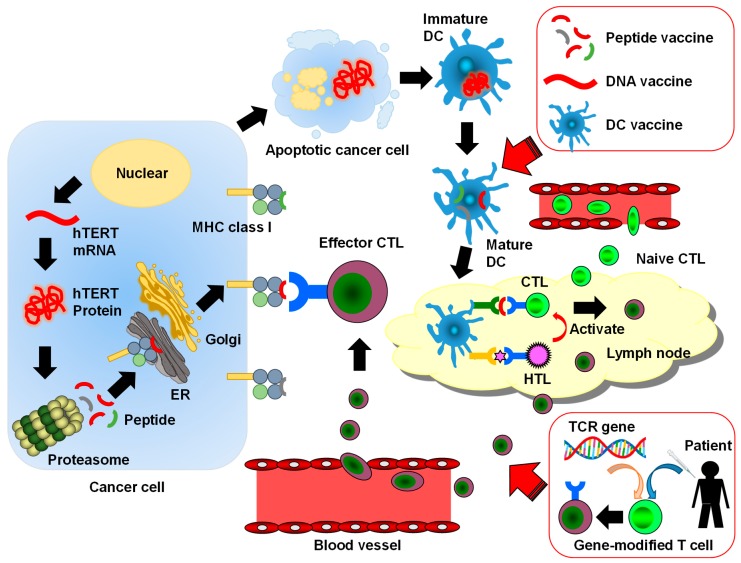
The human telomerase reverse transcriptase (hTERT)-specific cancer immune cycle and telomerase-targeted cancer immunotherapy: The hTERT protein produced in a cancer cell is cut to small peptides. The peptides are complexed with major histocompatibility complex (MHC) class I molecules and are presented on the cell surface for cytotoxic T lymphocytes (CTLs). Apoptotic cancer cells or proteins produced by cancer cells are phagocytosed by immature dendritic cell (DC), and the DCs present immunogenic hTERT-derived peptides to CTLs in a lymph node. The CTLs are recruited to a tumor site and kill cancer cells through the recognition of immunogenic peptides presented by cancer cells. Helper T (Th) cells stimulate CTLs and enhance their ability to kill cancer cells. ER means endoplasmic reticulum. The red arrows and boxes show the telomerase-targeted cancer immunotherapies. These therapies accelerate the cancer immune cycle.

**Table 1 ijms-20-01823-t001:** hTERT-derived immunogenic peptides.

Sequence *	Position	HLA Restriction	Immune Response for CD4/CD8	Year of Report	Refs.
MPRAPRCRA	1–9	HLA-B7	−/+	2006	[36]
RLGPQGWR	30–37	HLA-A2	−/+	2007	[33]
*R*LGPQGWRV	30–38	HLA-A2	−/+	2007	[33]
APSFRQVSCL	68–77	HLA-B7	−/+	2001	[41]
APSFRQVSCLKELVA	68–82	HLA-DR	+/−	2018	[29]
AYQVCGPPL	167–175	HLA-A24	−/+	2006	[35]
RPAEEATSL	277–285	HLA-B7	−/+	2006	[36]
VYAETKHFL	324–332	HLA-A24	−/+	2006	[35]
YLEPACAKY	325–333	HLA-A1	−/+	2005	[34]
RPSFLLSSL	342–350	HLA-B7	−/+	2006	[36]
RPSLTGARRL	351–360	HLA-B7	−/+	2006	[36]
YWQMRPLFLELLGNH	386–400	HLA-DP	+/−	2011	[39]
DPRRLVQLL	444–452	HLA-B7	−/+	2006	[37]
VYGFVRACL	461–469	HLA-A24	−/+	2006	[35]
FVRACLRRL	464–472	HLA-B7	−/+	2006	[37]
ILAKFLHWL	540–548	HLA-A2	−/+	2000	[30]
LAKFLHWLMSVYVVE	541–555	HLA-DP	+/−	2011	[38]
LLRSFFYN	555–563	HLA-A2	−/+	2007	[40]
RLFFYRKSV	572–580	HLA-A2	−/+	2002	[31]
*Y*LFFYRKSV	572–580	HLA-A2	−/+	2002	[32]
LFFYRKSVWSKLQSI	573–584	HLA-DP	+/−	2011	[38]
EARPALLTSRLRFIPK	611–626	HLA-DR,DQ,DP	+/−	2011	[38]
RPALLTSRLRFIPKP	613–627	HLA-DP	+/−	2011	[38]
DYVVGARTF	637–645	HLA-A24	−/+	2006	[35]
ALFSVLNYERARRPGLLGASVLGLDDIHRA	660–689	HLA-A2,DR	+/+	2011	[39]
SVLNYERARRPGLLG	663–677	HLA-DR	+/−	2011	[39]
RPGLLGASVLGLDDI	672–686	HLA-DR1,7,15	+/−	2002	[43]
PGLLGASVLGLDDIH	673–687	HLA-A2,DR	+/+	2011	[39]
GLLGASVLGL	674–683	HLA-A2	−/+	2011	[39]
LLGASVLGL	675–683	HLA-A2	−/+	2012	[44]
LTDLQPYMRQFVAHL	766–780	HLA-DR1,7,15	+/−	2003	[42]
CYGDMENKL	845–853	HLA-A24	−/+	2006	[35]
RLVDDFLLV	865–873	HLA-A2	−/+	2000	[30]
KLFGVLRLK	973–981	HLA-A2,A3	−/+	2001	[41]
DLQVNSLQTV	988–997	HLA-A2	−/+	2002	[32]
*Y*LQVNSLQTV	988–997	HLA-A2	−/+	2002	[32]
TYVPLLGSL	1088–1096	HLA-A24	−/+	2006	[35]
LPGTTLTAL	1107–1115	HLA-B7	−/+	2006	[37]
LPSDFKTIL	1123–1131	HLA-B7	−/+	2006	[37]

* The amino acids in italics and with an underline are mutated.

**Table 2 ijms-20-01823-t002:** The reported clinical trials of telomerase-targeted cancer vaccines.

Name	Clinical Trial Phase	Cancer Targeted	Clinical Response	Adverse Events	Year of Report	Ref.
GV1001	Phase II (combined with cyclophosphamide)	Hepatocellular carcinoma (HCC)	No clear GV1001-specific immune responses 17/40 SD	Well-tolerated	2010	[52]
	Phase I/II (combined with temozilomide)	Melanoma	Immune responses 5/25 PR, 6/25 SD	Well-tolerated	2011	[53]
	Phase I/II	Lung and colon cancer and melanoma	Immune responses	Well-tolerated	2012	[54]
	Phase I/II (combined with or without GM-CSF or gemcitabine)	Pancreatic cancer	Immune responses	Mild vaccination-related adverse events	2014	[55]
	Phase III (GV1001 with or without gemcitabine and capecitabine)	Pancreatic cancer	Adding GV1001 to chemotherapy did not improve the overall survival of patients.	No additional adverse events	2014	[56]
	Phase I/II (combined with hTERT_540_ peptides)	Non-small cell lung cancer (NSCLC)	Immune responses 7/26 SD (1/26CR after clinical trial)	Well-tolerated	2006	[57]
	Phase I	Melanoma	Immune responses	Well-tolerated	2011	[58]
UV1	Phase I/IIa	Prostate cancer	Immune responses 17/22 SD	Injection site pruritus	2017	[59]
Vx-001	Phase I/II	NSCLC	Immune responses 8/22 SD	Well-tolerated; Local skin reactions	2007	[49]
	Phase I/II (optimized Vx-001)	Breast cancer, colorectal cancer, head and neck cancer, HCC, melanoma, prostate cancer, kidney cancer, pancreatic cancer, cholangiocarcinoma, and others with advanced solid tumors, other than NSCLC	Immune responses 1/55 CR, 1/55 PR, 18/55 SD	Well-tolerated	2012	[50]
	Phase I/II (optimized Vx-001-TERT(572Y))	Chemo-resistant advanced solid tumors	Immune responses Better clinical outcome in responders than nonresponders	Well-tolerated	2012	[51]
	Phase II	NSCLC	Immune responses 3/46 PR, 13/46 SD	Well-tolerated	2014	[60]
Gx-301	Phase I/II	Prostate and renal cancer	Immune responses	Well-tolerated	2013	[61]
hTERT_461_	Phase I	HCC	Immune responses	No significant adverse events	2015	[62]
Dendritic cell vaccines	Phase I (Pulsed with hTERT_540_ peptide)	Breast and prostate cancer	Immune responses 4/7 SD	No significant adverse events	2004	[63]
	Phase I/II (Transfected with hTERT mRNA)	Prostate cancer	Immune responses; a reduction of PSA and molecular clearance of circulating micrometastases	Well-tolerated	2005	[64]
	Phase I/II (Pulsed with TERT_540_ peptide)	Prostate, breast, lung, colorectal, renal, head and neck cancer, and melanoma	Immune responses 4/16 PR	Well-tolerated; mild flu-like symptoms and fever	2009	[65]
	Phase I/II (GRNVAC1)	Acute myeloid leukemia	Immune responses Favorable disease-free survival	Well-tolerated; local transient erythema	2010	[66]
	Phase I/II (TAPCells vaccine)	Melanoma and prostate cancer	Immune responses	Well-tolerated	2013	[67]
	Phase I (DC pulsed with hTERT_572_, CEA and survivin-derived peptides.	Pancreatic cancer	Immune responses 4/8 SD	Well-tolerated Fatigue and self-limiting flu-like symptoms	2017	[68]

CEA, carcinoembryonic antigen; NSCLC, non-small cell lung cancer; HCC, hepatocellular carcinoma; PR, partial response; SD, stable disease; CR, complete response; PSA, prostate specific antigen.

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
