# Peer review of "Telomerase-Targeted Cancer Immunotherapy"

_ijms, 2019, doi:10.3390/ijms20081823_

Reviewer 1 Report

The manuscript contains a well written review on topic of cancer immunotherapy with reference to telomerase. As a general rule most of the cancers have upregulated telomerase that would make it an ideal target for immunotherapy. While authoprs have discussed various types of advances made on the subjected, they have not discussed the upregulation of telomerase due to TERT promoter mutations and implication of increased telomerase levels due to those mutations as discussed in a review by Maurizio Zanetti in Nature Reviews Clinical Oncology 2017. The should also cite papers that described discovery of TERT promoter mutations in Science (2013).

Author Response

Reviewer 1

Comments and Suggestions for Authors

The manuscript contains a well written review on topic of cancer immunotherapy with reference to telomerase. As a general rule most of the cancers have upregulated telomerase that would make it an ideal target for immunotherapy. While authoprs have discussed various types of advances made on the subjected, they have not discussed the upregulation of telomerase due to TERT promoter mutations and implication of increased telomerase levels due to those mutations as discussed in a review by Maurizio Zanetti in Nature Reviews Clinical Oncology 2017. The should also cite papers that described discovery of TERT promoter mutations in Science (2013).

This comment is very constructive. We cited papers that described discovery of TERT promoter mutations in Science (2013) and written by Zanetti et al. in Nature Reviews Clinical Oncology (2017).

We added sentences in the introduction as follows.

“In addition, recent studies have shown mutations exist frequently in the promoter region of hTERT gene and increase activity of telomerase to play a major role in tumorigenesis [8,9]. These findings have triggered researchers to reconsider the usefulness of treatments targeting hTERT [10].” (Page 3, Line 44)

Reviewer 2 Report

In the manuscript "Telomerase-targeted Cancer Immunotherapy" the authors summarized the current literature related to the immunotherapy strategies that target hTERT for cancer treatment.

The review is generally well organized and comprehensive in the proposed issue; however, there are some aspects that should be included to improve and clarify the manuscript:

- a significant criticism would be that the paper is a list of the current literature on immunotherapeutic strategies that target hTERT, but it lacks an in-depth discussion of the presented data and of what might be the best therapeutic approach.

- the authors describe the advantages derived from immunotherapy targeting hTERT for cancer treatment, but antitelomerase therapy may favor the emergence of adaptive responses, such as the activation of the alternative lengthening of telomeres (ALT) mechanism (Hu J, et al. Cell 2012;148:651-63); these considerations should be included and discussed.

- for better understanding, the authors should provide more detail on how each immunotherapeutic strategy works, with advantages and disadvantages   

 Author Response

Reviewer 2

Comments and Suggestions for Authors

In the manuscript "Telomerase-targeted Cancer Immunotherapy" the authors summarized the current literature related to the immunotherapy strategies that target hTERT for cancer treatment.

 The review is generally well organized and comprehensive in the proposed issue; however, there are some aspects that should be included to improve and clarify the manuscript:

 - a significant criticism would be that the paper is a list of the current literature on immunotherapeutic strategies that target hTERT, but it lacks an in-depth discussion of the presented data and of what might be the best therapeutic approach.

 - the authors describe the advantages derived from immunotherapy targeting hTERT for cancer treatment, but antitelomerase therapy may favor the emergence of adaptive responses, such as the activation of the alternative lengthening of telomeres (ALT) mechanism (Hu J, et al. Cell 2012;148:651-63); these considerations should be included and discussed.

 - for better understanding, the authors should provide more detail on how each immunotherapeutic strategy works, with advantages and disadvantages  

 Responses to the comments of reviewer 2

 This comment is very constructive. We added the sentences to mention more detail on how each immunotherapeutic strategy works and our opinion to describe the best immunotherapeutic approach as a Telomerase-targeted Cancer Immunotherapy. We also added the sentences to describe antitelomerase therapy and cited papers by Hu J, et al. in Cell.

In addition, we changed Figure 1 and the figure legend for better understanding how each immunotherapeutic strategy works.

 “Many immunotherapies using hTERT-derived peptides, DNAs and DCs have been developed, but their effects so far are modest. The reason is that hTERT is a self-antigen, and T cells exerting an effect against such self-antigens are hard to induce in vivo. In addition, the TCR affinities of the induced T cells are low and therefore the antitumor effect of these T cells might be weak. Furthermore, the expression of anti-tumor effects by these vaccines usually need some time and such antitelomerase therapy may favor the emergence of adaptive responses, such as the activation of the alternative lengthening of telomeres mechanism reported by Hu et al. [103]. In order to overcome such a point, it is important to administer a large amount of T cells having a TCR capable of exerting a certain antitumor effect.

As described above, in the field of cancer immunotherapy, treatment with TCR gene-modified T cells has been attempted in many cancer types. In immunotherapy targeting hTERT developed so far, this therapeutic method using genetically modified T cells is a mechanism to administer T cells with TCRs that can reliably exert antitumor effects in vivo and it is the most promising treatment in that a reliable antitumor effect can be obtained in a short time. On the other hand, since it is thought that an immunosuppression mechanism via immune checkpoint molecules such as PD-1 is considered to be working at tumor locality, combined therapy with immune checkpoint inhibitors will be further promoted and seems to be effective.” (Page 21, Line 331)

 Figure 1.hTERT specific cancer immune cycle and telomerase-targeted cancer immunotherapy. hTERT protein produced in cancer cell is cut to small peptides. The peptides are complexed with major histocompatibility complex (MHC) class I molecules and are presented on the cell surface for cytotoxic T lymphocytes (CTLs). Apoptotic cancer cells or proteins produced by cancer cells are phagocytosed by immature dendritic cell (DC) and the DCs present immunogenic hTERT-derived peptides to CTLs in lymph node. The CTLs are recruited to tumor site and kill cancer cells through the recognition of immunogenic peptides presented by cancer cells. Helper T (Th) cells stimulate CTLs and enhance their ability of cancer cell killing. ER means endoplasmic reticulum. Red arrows and boxes show telomerase-targeted cancer immunotherapies. These therapies accelerate the cancer immune cycle.

Reviewer 3 Report

In this review article, the authors summarized the recent advance in TERT-based anti-cancer immunotherapy, which is clear and well written.

One issue that needs to be addressed is if TERT-based immunotherapy leads to killing each other of host immune cells, thereby impairing host immune system and affecting therapeutic efficacy, because both B and T lymphocytes are telomerase/TERT-proficient cells, especially activated ones.

Author Response

Reviewer 3

Comments and Suggestions for Authors

In this review article, the authors summarized the recent advance in TERT-based anti-cancer immunotherapy, which is clear and well written.

 One issue that needs to be addressed is if TERT-based immunotherapy leads to killing each other of host immune cells, thereby impairing host immune system and affecting therapeutic efficacy, because both B and T lymphocytes are telomerase/TERT-proficient cells, especially activated ones.

Responses to the comments of reviewer 3

 This comment is very important and constructive. As the reviewer suggests, hematopoietic progenitor cells and both B and T lymphocytes have high telomerase activity. This means that TERT-based anti-cancer immunotherapy not only kill cancer cells but also these lymphocytes with high telomerase activity. In previous studies with hTERT-derived peptide vaccine and dendritic cells, no serious adverse events regarding lymphopenia have been reported. However, it should be carefully observed in future studies using hTERT-targeted TCR engineered T cell therapy.

 According to the reviewer’s suggestion, we added the following sentences in the text.

 “As mentioned above, severe adverse events have not been observed in hTERT-targeted immunotherapy. However, in this immunotherapy, it might be necessary to be careful about abnormalities in the host’s immune system. Hematopoietic progenitor cells and both B and T lymphocytes have high telomerase activity. This means that hTERT-based anti-cancer immunotherapy not only kill cancer cells but also these lymphocytes with high telomerase activity. In previous studies with hTERT-derived peptide vaccine and dendritic cells, no serious adverse events regarding lymphopenia have been reported (Table 2). However, it should be carefully observed in future studies using new treatment strategies described later.” (Pgae 18, Line 265)

Round  2

Reviewer 1 Report

No further comments

Reviewer 2 Report

the revision slightly improved the manuscript